# Frother Characterization Using a Novel Bubble Size Measurement Technique

Junyu Wang [1], Gordon Forbes [1] and Elizaveta Forbes [1,2,*]

1 Julius Kruttschnitt Mineral Research Centre, University of Queensland, Brisbane 4068, Australia; j.wang30@uq.edu.au (J.W.); g.forbes@uq.edu.au (G.F.)
2 ARC Centre of Excellence for Eco-Efficient Beneficiation of Minerals (CE200100009), Shortland 2307, Australia
* Correspondence: l.forbes@uq.edu.au

**Abstract:** Bubble size measurement is a vital part of flotation system analysis and diagnostics. This work evaluates a commercial camera probe as a novel method for in situ bubble size measurement. This device is compared to the conventional Anglo Platinum Bubble Sizer (Stone Three™). It was found that, in laboratory applications, the in situ bubble size analysis technology appears to be a more user-friendly and reliable option for determining bubble size in flotation, whereas the Anglo Platinum Bubble Sizer is more applicable for full scale industrial work. This probe was then used to conduct a rigorous comparison of the behavior of different frother chemistries at a variety of background solution ionic strength conditions. The critical coalescence concentrations and the minimum Sauter mean bubble diameters were determined. Five frothers were compared in terms of their ability to reduce bubble size and sensitivity to salinity. In order to adjust plant recipe and control strategy accordingly, it is recommended that the plant would need to use less frother during periods of the high salinity of process water to achieve the minimum Sauter mean bubble size.

**Keywords:** PVM probe; APBS; bubble size measurement; frother characterization; coal flotation

## 1. Introduction

### 1.1. Bubble Size in Flotation

In mineral flotation, the surfaces of bubbles are the primary vehicles for the recovery of valuable mineral particles. The available bubble surface area is largely determined by the size of the individual bubbles, which makes bubble size one of the key factors affecting flotation recovery [1–3].

Parameters such as Sauter mean bubble diameter ($d_{32}$) and bubble surface area flux ($S_b$) are also essential components of determining flotation rate constants in models predicting flotation behavior [4–6]. For this reason, the accurate measurement of bubble size distributions has been an essential requirement of flotation cell characterization and performance assessment.

### 1.2. Frothers and Their Characterization

One of the major functions that frother reagents have in froth flotation is to minimize bubble size through retarding coalescence. The surface area of bubbles is the main vehicle for the recovery of particles in the pulp phase. For this reason, the size of the bubbles should be as small as possible to maximize the bubble surface area flux [3].

The strength and efficacy of the frother reagent are typically evaluated using the CCC (Critical Coalescence Concentration). The CCC determination is based on the relationship between bubble size and frother concentration. Thus, accurate bubble size measurement is essential for accurate assessments of frothers for industrial use [7].

Frother efficiency is affected by solution properties [8]. In industrial practice, the ion content of process water can vary greatly between operations, but also as a function

of seasonal change within the same operation. Figure 1 shows the variation in electrical conductivity of the dam water at a coal mine in Queensland. Electrical conductivity is a useful proxy for the overall ion content of water [9]. As can be seen, the electrical conductivity of the process water varies significantly between 4 and 13 mS/cm. This coal mine adds frother into its Jameson flotation cells. However, excess frother use might eventually destabilize the froth and lead to a slower flotation rate [10]. Therefore, it's important that the minimum amount of frother is added, such that the critical coalescence concentration can be achieved without adversely affecting the flotation performance.

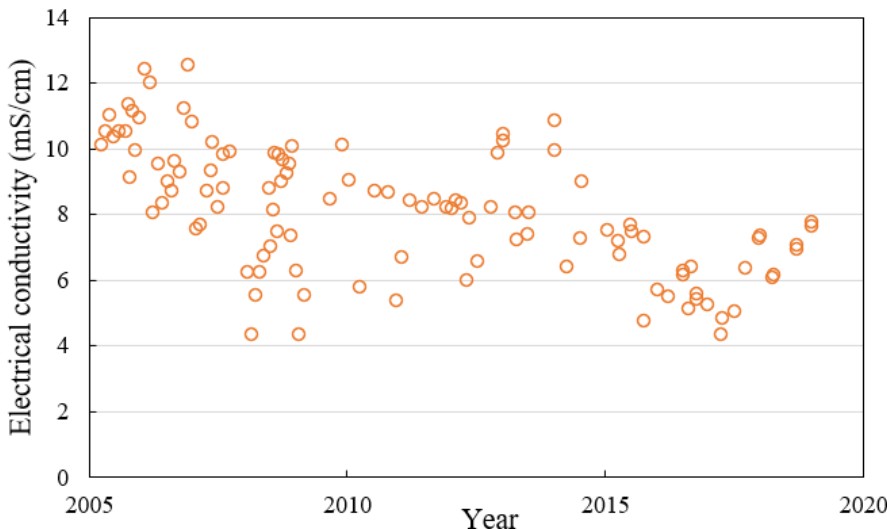

**Figure 1.** Electrical conductivity of the dam process water at a coal mine in Queensland, Australia.

### 1.3. Traditional Bubble Size Measurement Tools

Bubble size measurement devices have been a staple of flotation analysis toolkits for well over 40 years. In their technical note on bubble size measurement, Chen et al. provide an extensive list of different bubble size analysis techniques dating back to 1970 [7]. One of the most enduring and popular methods is that of bubble image capture (via either still photography or video recording), followed by image analysis and bubble segmentation.

While the early bubble size measurement devices were developed for laboratory use, the technique gained prominence when devices capable of measurements in industrial flotation cells became available. One of the more frequently utilized devices is the Anglo Platinum Bubble Sizer (APBS). The device design was developed by the University of Cape Town in collaboration with Anglo Platinum as a modification of a device built by McGill University [7,8], and currently manufactured by Stone Three.

Although there are many different individual bubble viewers and bubble size analyzers available, the basic design of these instruments has remained largely the same. These devices all tend to share the following features [7–9,11,12]:

- Bubbles are sampled from the pulp through a bubble rise tube or column, leaving the solids behind.
- The bubbles rise through the sampling column into a measurement chamber. This chamber can be a bubble viewer window, where bubble images are captured employing a camera. Alternatively, bubbles can be captured by a bell capillary, where the volume of individual bubbles is calculated from the total volume of collected gas.
- In all cases, the measured size of the bubbles needs to be adjusted to compensate for a difference in hydrostatic pressure between the sampling point and the measurement point.
- Before measurement, the instrument needs to be filled with water that matches the chemical composition of the flotation pulp. However, an additional frother is usually added to the analyzer to prevent bubble coalescence inside the sampling tube.

- During the measurement, the water contained within the instrument is displaced by the collected gas. The downward flow of water then flows counter-current to the rising bubbles.

While these methods of bubble size measurement are undoubtedly effective, they still have disadvantages. The most obvious of these is the potential inconsistency in bubble size between the sampling point and the measurement point. The downward flow of water from the measurement chamber has the potential to bias the bubble size distribution, as the smallest bubbles will be swept out of the tube by the flow. An additional concern is the possibility of bubble coalescence within the riser tube. For this reason, extra frother reagent is typically added to the measurement chamber at a concentration that exceeds the Critical Coalescence Concentration (CCC) of the frother reagent [13,14].

Until recently, there were no bubble size measurement tools available that could operate in situ within the flotation pulp. This also means that there was no effective way to validate the accuracy of traditional bubble size measurement devices. However, due to recent advances in in situ measurement systems, such tools are becoming increasingly available.

*1.4. In Situ Measurement Tools*

In situ measurement instrumentation such as the Focused Beam Reflectance Measurement (FBRM) and ParticleView ™ Measurement (PVM) probes (both manufactured by Mettler Toledo) have previously been used to characterize flotation systems [15,16]. Such devices typically consist of a sensor mounted at the end of a water-proof tube that is inserted directly into the fluid where the measurement takes place. However, the use of these techniques has largely focused on the analysis of mineral particles and particle aggregates within the flotation pulp. Indeed, such devices are specifically manufactured to study very fine entities such as fine particles and emulsion droplets. The use of a tool such as the PVM probe to analyze the size of bubbles in a flotation system has been attempted at CSIRO Minerals, but the results have not been made available in published literature. There is currently a probe under development by Stone Three that is capable of measuring in situ bubble size in industrial flotation cells. However, this probe is not available on the market yet.

The objectives of the current study are thus to evaluate the novel size measurement technique in spherical bubble detection applications and to demonstrate the use of the novel technique in an industrial context.

To evaluate the novel bubble size measurement technique, bubble size was measured and compared with the traditional method of using the Anglo Platinum Bubble Sizer (APBS). The capabilities and limitations, the convenience for lab and site installation, and the time requirement of each technique were compared and discussed.

An assessment of five frother candidates was evaluated in the current study for a coal mine in Queensland, Australia, at the time of its frother contract being up for renewal. This coal mine also experienced high levels of variability in the salinity of its process water. A rigorous comparison of the behavior of potential frother candidates at a variety of background solution ionic strength conditions was required for the purpose of creating a shortlist of reagents for full-scale industrial trials. This work specifically looked at determining the critical coalescence concentration (CCC) and characteristic two-phase bubble size at CCC for each formulation and as a function of process water salinity.

## 2. Materials and Experimental Methods

In this section, the experimental setup, materials used for the tests, and experimental methods are introduced.

*2.1. Bubble Generation Systems*

2.1.1. Bubble Generation Columns

Two columns were used for bubble generation. The first experimental program was to compare the two bubble size measurement techniques, PVM and APBS probes. A 5 L rectangular Perspex column with 50 cm in height and $10 \times 10$ cm$^2$ cross-sectional area, with a sparger sitting at the bottom, was used for this purpose. This was because the APBS was 1.5 m in height and the sampling tube added another 0.7 m to the total height of the rig, which would not allow for the use of a mechanical cell sitting on a bench. Instead, this column did not have mechanical agitation on its bottom and was placed on the floor, which enabled the setup of the APBS on top of it. The air feed rate was set at 3 L/min, equivalent to a superficial gas velocity ($J_g$) of 0.5 cm/s. In all cases, the bubble size was measured at a fixed depth of 25 cm below the cell lip.

The second program was to use the PVM probe to characterize frothers in a two-phase system. The small scale of this probe provided more flexibility of testing. A bottom-driven column, 23 L with a height of 58.3 cm and $20 \times 20$ cm$^2$ cross-sectional area, was used for this purpose. Air was supplied and dispersed by a speed-controlled impeller fitted at the bottom of the cell. The air rate was set at 6 L/min, equivalent to a superficial gas velocity ($J_g$) of 0.25 cm/s. The agitation speed was set at 1200 RPM.

2.1.2. Frothers

The current frother (A, provided by the mine site) was compared with a conventionally used frother (MIBC, supplied by Rowe Scientific Pty. Ltd., Doveton, Australia) and three other candidate frothers (commercial reagents labelled B, C, and D). Each frother was evaluated within the 0 to 100 ppm concentration range. Details of the five frothers under study are given in Table 1.

**Table 1.** Compositions of frothers under the study.

| Frother Nomenclature | Composition/Ingredients |
| --- | --- |
| A | Not disclosed |
| MIBC | Methyl isobutyl carbinol |
| B | Oxygenated hydrocarbons |
| C | Blend of hydrocarbons and glycol |
| D | Blend of hydrocarbons |

2.1.3. Electrolytes

The background solution used for the tests was adjusted to match the ion content of the site operation. The electrolyte composition of the solution is summarized in Table 2. The solution pH was adjusted and maintained at 8.00 (7.90–8.10) using 1M Sodium Hydroxide (NaOH) solution. All the reagents were sourced from Rowe Scientific Pty Ltd.

**Table 2.** Composition of synthetic process water.

| Salts | Concentration (mg/L) | | |
| --- | --- | --- | --- |
| | Low | Medium | High |
| $CaCl_2 \cdot 2H_2O$ | 114 | 220 | 477 |
| $MgCl_2$ | 67 | 204 | 389 |
| KCl | 12 | 21 | 74 |
| NaCl | 2481 | 4194 | 5555 |

The conductivity of the synthetic water was measured, and the ionic strength (IS) was calculated using Equation (1).

$$IS = \frac{1}{2} \sum C_i Z_i^2 \tag{1}$$

where $C_i$ is the molar concentration of the ions and $Z_i$ is the ion charge density.

Figure 2 presents the comparison between the calculated ionic strengths and measured conductivities of the site process water and the synthetic water used in the current test work. The data shows that the synthetic water of variant salt concentration covered the same ionic strength range, but showed a slightly lower conductivity as compared with plant water. Additionally, in both cases, the relationship between conductivity and ionic strength is linear, suggesting that conductivity is an excellent proxy for salinity levels.

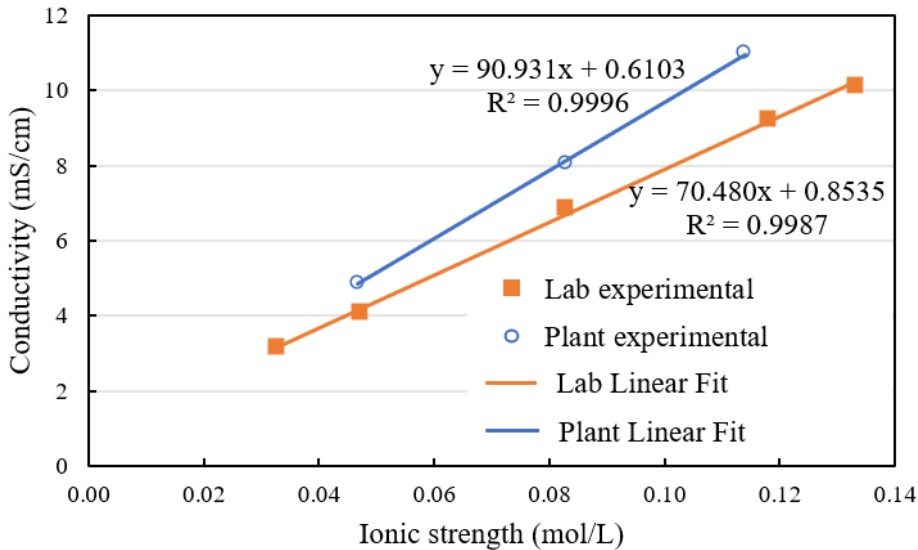

**Figure 2.** Comparison of lab and plant process water conductivity.

2.1.4. Experimental Procedure

Salts needed to make synthetic process water at a specific concentration level were measured and dissolved into de-ionized water. After it was transferred to the cell, extra de-ionized water was added to reach the target of 20 L total volume. At the same time, the air was turned on at the specified air rate. The system was then conditioned for 5 min, and pH was adjusted to 8 during this process. Frother was then added at the required dosage level. The system was conditioned for a further 1 min before the test began. Images were taken with using either one of the two tools introduced below.

*2.2. Bubble Size Measurement Techniques*

2.2.1. APBS Probe

The Anglo Platinum Bubble Sizer (APBS) probe was manufactured by Stone Three. The probe consists of a sampling tube 50 mm in diameter and 70 cm in length. The tube leads to a measurement chamber, where the bubbles are directed towards an inclined plane where their images are captured with a digital camera for subsequent processing. The bubble segmentation is performed via the dedicated software package APBS V2.3.1, also provided by Stone Three. The bubble sizer device, as well as an example of the bubble images both prior and post segmentation, are presented in Figure 3.

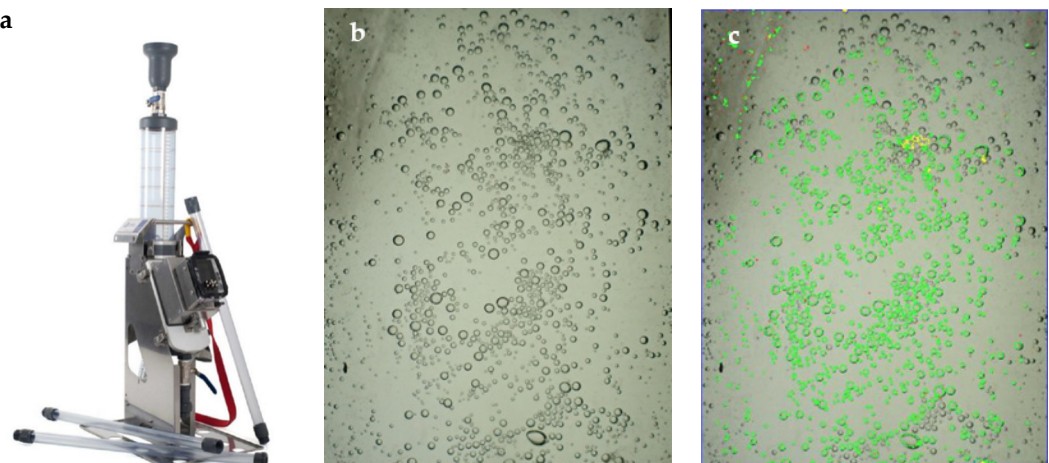

**Figure 3.** (**a**) APBS probe, (**b**) captured bubble images and (**c**) bubble segmentation, image size 91 × 69 mm.

Once the bubble size measurements were obtained, they were adjusted to correct for the difference in hydrostatic pressure between the sampling point and the bubble viewer box following the APBS operating manual, as per Equations (2) and (3).

$$d_1 = d_2 \cdot \sqrt[3]{\frac{P_2}{P_1}} \tag{2}$$

$$P_i = P_{atm} + \rho g h_i \cdot (1 - \varepsilon_g) \tag{3}$$

$d_i$: Bubble diameter at point $i$ (mm);
$P_i$: Pressure at point $i$ (Pa);
$P_{atm}$ : Atmospheric pressure (Pa);
$h_i$: Liquid reference height at point $i$ (m);
$\varepsilon_g$: Gas holdup (%);
$\rho$: Fluid density (kg/m$^3$);
$g$: gravitational acceleration (m/s$^2$).

2.2.2. PVM Probe

The in situ bubble size measurement was performed using the ParticleView$^{TM}$ V19 (Mettler Toledo) system with auto-lighting, which is illustrated in Figure 4a. The end of the probe containing the camera was positioned at 25 cm below the cell lip.

The probe camera has a field of view of 1300 × 890 μm. Inline images were automatically viewed and recorded using a software iC PVM$^{TM}$, which allows subsequent offline bubble size image analysis. Images were taken during tests at a rate of approximately two frames per second, collecting circa 800 images per test, as illustrated in Figure 4b,c.

Although the PVM probe comes with a dedicated image analysis software package, the software was unable to adequately detect large objects such as bubbles, as the PVM probe was not designed for bubble detection. Therefore, a dedicated bubble segmentation algorithm had to be written for this system.

The images were analyzed offline using standard image pre-processing techniques followed by the circular Hough transform. The Hough transform is typically used to identify circles with either a specific radius or a narrow range of radii [17,18], and was previously used by researchers for bubble detection application in flotation columns [19]. MATLAB's implementation allows for a search with a fairly large range of radii, but this was not suitable for some extreme cases where both large and small bubbles were present in the same image. In this case, the edge-threshold and sensitivity parameters were re-

adjusted to detect small and large bubbles and bubbles with clear and blurry boundaries, respectively. The bubble segmentation sequence is illustrated in Figure 5.

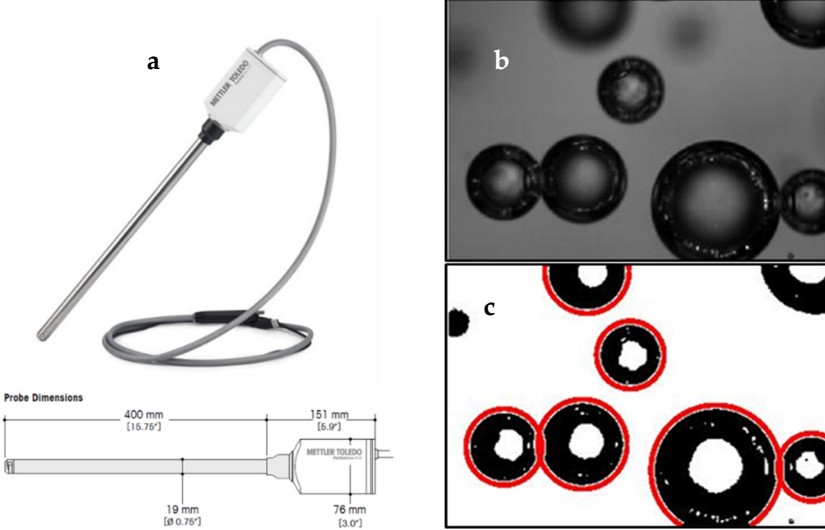

**Figure 4.** (**a**) PVM probe, (**b**) captured bubble images and (**c**) bubble segmentation, image size $1.30 \times 0.89$ mm.

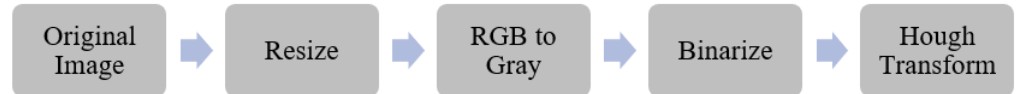

**Figure 5.** Image processing sequence.

### 2.2.3. Lab-Scale Experimental Set-Up

The lab-scale experimental set-up is shown in Figure 6. As can be seen, the installation of APBS requires more space, while PVM probe is more portable as compared with each other.

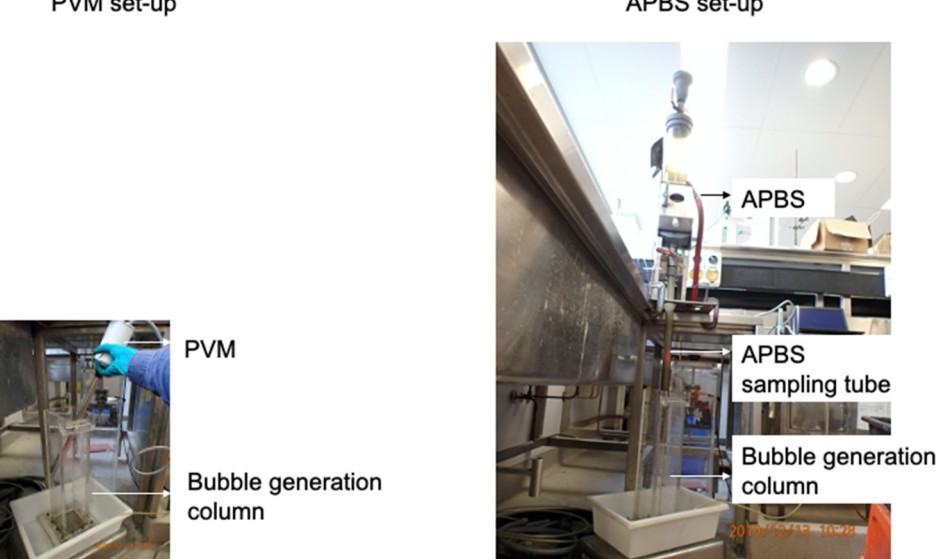

**Figure 6.** Lab-scale experimental set-up (different size images presented to enable the size comparison between the two sets of equipment).

### 2.3. Data Analysis

When reporting bubble sizes, one of the most common metrics is the Sauter mean bubble diameter, which is calculated based on the volume to surface ratio of the bubbles, as illustrated in Equation (4) [20,21].

$$d_{32} = \frac{\sum_{i=1}^{n} d_i^3}{\sum_{i=1}^{n} d_i^2} \tag{4}$$

$d_{32}$: $d_{32}$ Sauter meanbubble diameter (mm);
$d_i$: $i$th bubble diameter (mm);
$n$: Number of bubbles.

The Critical Coalescence Concentration (CCC), which was defined as a concentration where bubbles cease to coalesce [13], was then determined. The method for the determination of CCC is graphically demonstrated in the result section.

## 3. Results

### 3.1. Comparison of Novel and Conventional Bubble Size Measurement Techniques

Bubble size measurement tests using the two different techniques were carried out in a background solution of medium salinity (with an ionic strength of 0.083 mol/L and a conductivity of 6.85 mS/cm) with the addition of 25 ppm MIBC. A comparison of the two techniques is presented in Table 3.

**Table 3.** Comparison of the PVM and APBS bubble size measurement techniques.

| Comparison | PVM | APBS |
|---|---|---|
| Application field | Micro-field (a few μm–1.3 mm) | mm-cm-field (200 μm–a few cm) |
| Size | 0.55 m–height 0.076 m–diameter | 2.00 m–height ~0.300 m–diameter |
| Weight | <1 kg | ~10 kg |
| Lab installation | Easy | Difficult |
| Picture taken point | In situ | 1 m above sampling point |
| Time requirements per test | ~30 min | ~20 min |

As can be seen from the table, PVM provides more details at the micron level, while APBS's field of view is in the millimeter range. Specifically, APBS has a field of view of 68.5 mm by 91 mm, which allows the capture of millimeter-to-centimeter-sized bubbles, as illustrated by Figure 7b. On the contrary, the PVM probe has a much smaller field of view, 0.89 mm × 1.3 mm, which allows the capture of only small bubbles in the micro-sized range, as shown in Figure 7a.

The smallest bubble size obtained with the APBS probe is 207 μm in size, while the PVM probe detects a significant quantity of bubbles as fine as 20 μm, as shown in Figure 8.

One of the possible reasons for the lack of detection of fine bubbles by the APBS probe is that the segmentation algorithm is not optimized for such fine bubbles. The resolution of the image and subsequent image segmentation is too low to tell if this is the case. APBS is a commercial system with its own software developed, so we cannot just replace with a better camera resolution or lens. Another possibility is that the very fine bubbles are not reaching the measurement chamber at all. They are likely being prevented from entering the sampling tube by the downward flow of water during the sampling process.

Conversely, the APBS probe picks up several bubbles in the range between 2500 and 3300 μm in size, while the largest bubble detected by the PVM probe was 1800 μm. This is most likely due to the small field of view of the PVM camera (1.17 mm$^2$), which is smaller than the size of the largest measured bubble. The Hough transform algorithm can

determine the size of a partially visible bubble, but reaches its limit at bubbles larger than the maximum detected size.

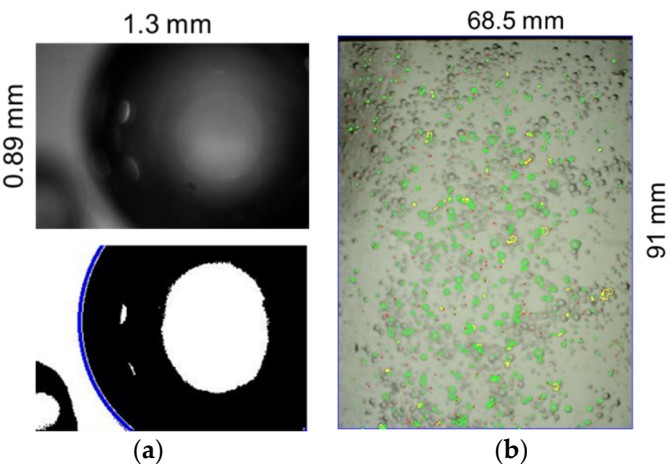

**Figure 7.** Field of view of PVM (**a**) and APBS (**b**) cameras.

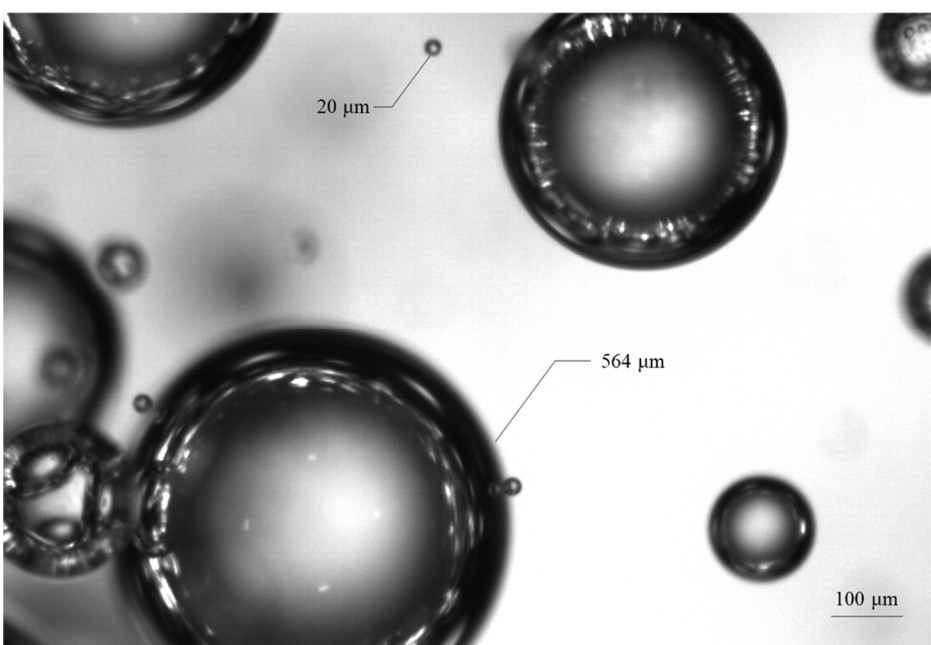

**Figure 8.** Bubbles captured with PVM probe.

As a result, the PVM probe gave a slightly smaller bubble size than the APBS probe does. Figure 9 shows the bubble size distributions obtained using the two techniques. Error bars represent 95% confidence intervals obtained from three experimental repeats. As can be seen, there was 8% of the total bubbles captured with the PVM probe that were smaller than 177 μm and were missing from the APBS curve. Conversely, 0.57% of bubbles captured with the APBS probe were larger than 2 mm and were not detected with the PVM probe. This small proportion of large bubbles affected the Sauter mean bubble diameter calculation.

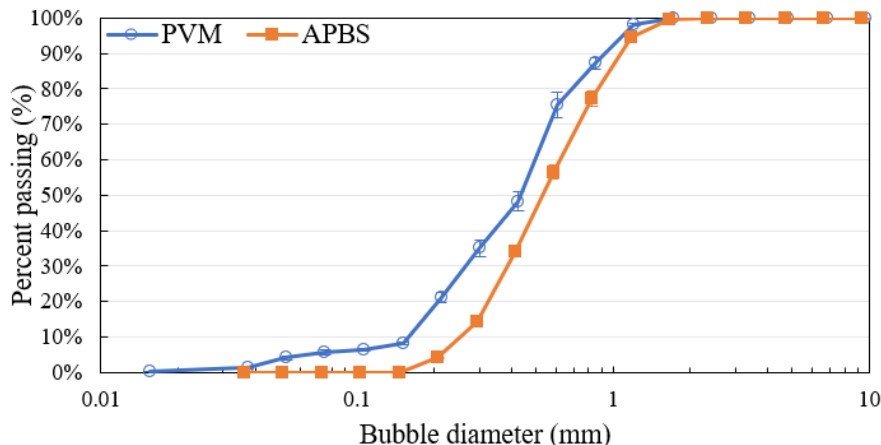

**Figure 9.** Bubble size distributions obtained using the PVM and APBS probes (there are error bars for APBS, but they are small and not visible).

Table 4 compares the Sauter mean bubble diameter obtained using the two techniques. The use of the PVM probe gave a smaller Sauter mean bubble diameter than APBS (after pressure correction), but the difference was not large (147 µm difference).

**Table 4.** Sauter mean bubble diameter measured with PVM and APBS probes.

| D32 (mm) | PVM | APBS |
|---|---|---|
| Test 1 | 0.954 | 1.058 |
| Test 2 | 0.945 | 1.125 |
| Test 3 | 0.988 | 1.142 |
| Average | 0.962 | 1.109 |
| 95% CI | 0.026 | 0.050 |

A summary of the measurement parameters for the two techniques is presented in Table 5. The first major difference is the number of images analyzed as part of the measurement and the consequent number of bubbles detected during the measurement.

**Table 5.** Summary of the measurement parameters for PVM and APBS probes.

| Comparison | PVM | APBS |
|---|---|---|
| Number of analyzed images | 2450 | 90 |
| Number of detected bubbles | 2894 | 83,683 |
| Test time (min) | 3 | 30 |
| Image analysis time (min) | 90 | 6 |
| Data post-processing time (min) | 0 | 30 |

In the case of the APBS probe, the image analysis field of view was approximately 63 cm$^2$, with each image containing about 1000 bubbles. In the case of the PVM probe, the field of view was only 1.17 mm$^2$. In such a small image area, the number of bubbles per image varied between 8–10 bubbles per image, to only a partially visible single bubble.

Overall, 90 images were obtained using the APBS probe (30 images per test run), containing over 80,000 bubbles. On the other hand, the PVM technique analyzed 2450 images, which yielded the detection of under 3000 bubbles.

Despite the 28-fold increase in the number of images processed, the time requirements for the use of the PVM probe technique were very similar to that of the APBS probe. The

time gains made in analyzing fewer images were lost in the labor-intensive process of probe installation and cleaning. In addition, the APBS probe required post-processing of the data to perform the bubble diameter correction due to the difference in hydrostatic pressure between the sampling point and the measurement point.

### 3.2. Use of PVM Probe for Two-Phase Frother Characterization

#### 3.2.1. Example of CCC Determination

Figure 10 shows the measured Sauter mean bubble diameter as a function of frother dosage that was obtained using Frother C at the low ionic strength (salinity) condition, as well as the graphical method of determining the CCC. The CCC values were obtained at the intersection of the sloped line approximating the curve at low concentration with the horizontal asymptote to the curve at high concentration [22].

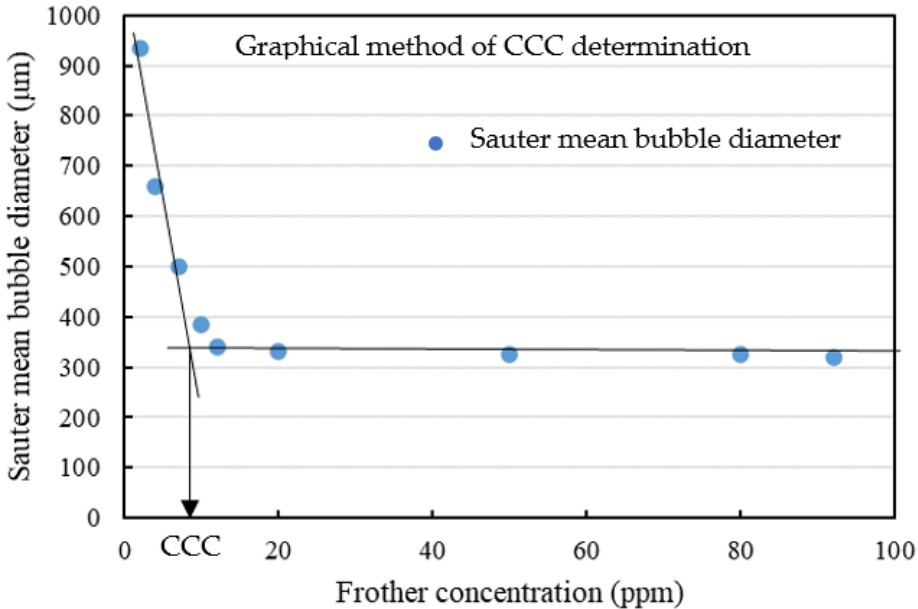

**Figure 10.** Example of CCC determination.

#### 3.2.2. Effect of Water Salinity on Bubble Size-Reduction

Figure 11 shows the Sauter mean bubble diameter as a function of the concentration of Frother A under three salinity level conditions. As expected, the Sauter mean bubble diameter decreased with increasing frother concentration and then reached a minimum at concentrations above CCC. At the same time, the results also clearly highlight the influence of salts on the frothing properties of an aqueous solution. As shown, smaller bubbles were generated using high salinity process water at the same frother concentration level and reached a minimum bubble size at lower frother concentrations than the case when low salinity process water is used.

This implies that the frother dosage required to reach a minimum bubble size in a specific system could be reduced at the condition of increased background water salinity. The presence of ions in the water is well known to have an effect on its frothing properties [15,16]. This synergy between frother dosage and pulp salinity could potentially be exploited to reduce the overall frother requirement at the flotation plant.

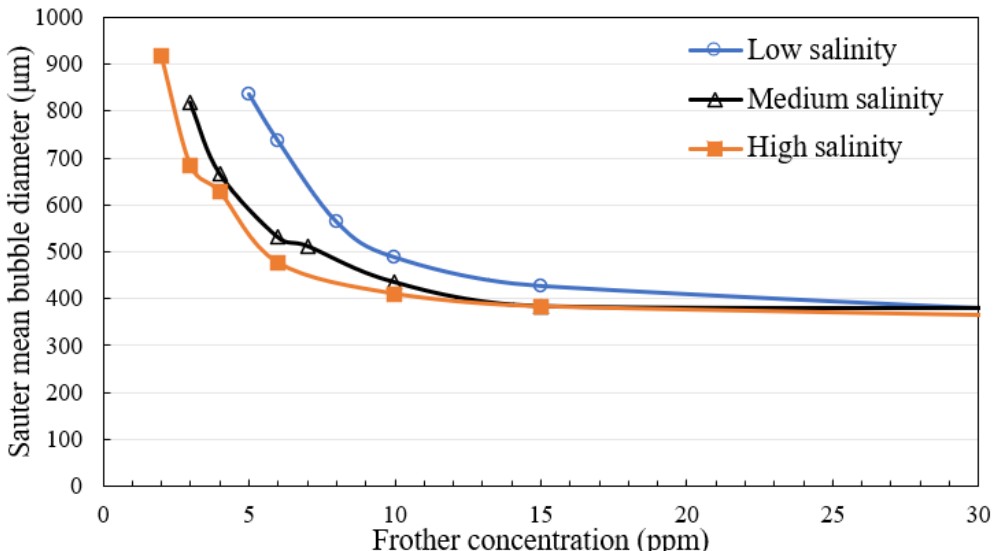

**Figure 11.** CCC comparison of Frother A at three water salinities.

### 3.2.3. CCC Comparison

Bubble size measurement was carried out for the five frothers at progressively increasing dosage and various background water salinity levels. The Sauter mean bubble diameter was then calculated at each condition and plotted against frother concentration. CCC of each frother at various water quality conditions was then determined graphically, as per the method introduced in Section 3.2.1, and was compared. The results are shown in Figure 12.

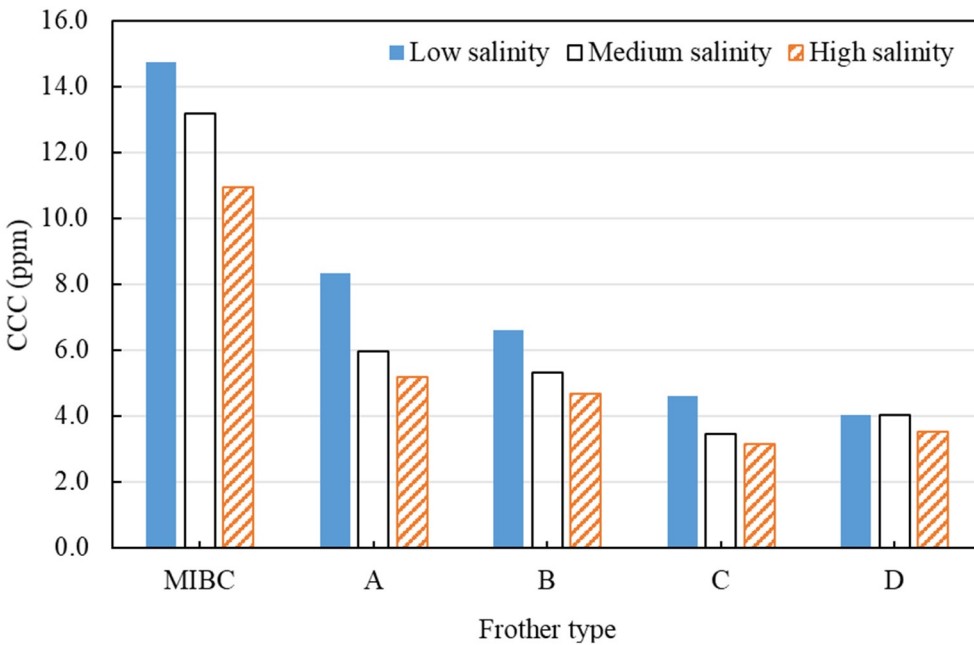

**Figure 12.** CCC comparison of five frothers at three water salinities.

MIBC was found to have the highest CCC out of the tested frothers, ranging between 12 and 15 ppm, which suggests that higher concentration is required for MIBC to effectively reduce bubble size to a minimum. The remaining four frothers showed a much lower CCC (5–12 ppm), suggesting they can reduce bubble size to a minimum at lower dosages than MIBC. Take Frother D as an example, the dosage can be potentially reduced by half compared to MIBC to effectively reduce bubble size to a minimum.

Apart from the type of frothers, it can be seen that water quality, in this case, the salinity or the ionic strength of process water, also affects the frothing properties of frothers. When frother is weak, the effect of salinity is more prominent. When frother is very strong, the additional effect of ions on frothing properties is less visible. It is recommended that the plant would need to use less frother during periods of the high salinity of process water to achieve the minimum Sauter mean bubble size. Frother D was the most indifferent to water salinity levels.

### 3.2.4. Minimum Sauter Mean Bubble Diameter at CCC

In addition to determining the CCC for each frother formulation as a function of water salinity, the minimum Sauter mean bubble diameter was also determined. The results are shown in Figure 13, which represents the average of three bubble size readings at frother concentrations above the CCC. The error bars represent the 95% confidence intervals of the mean values.

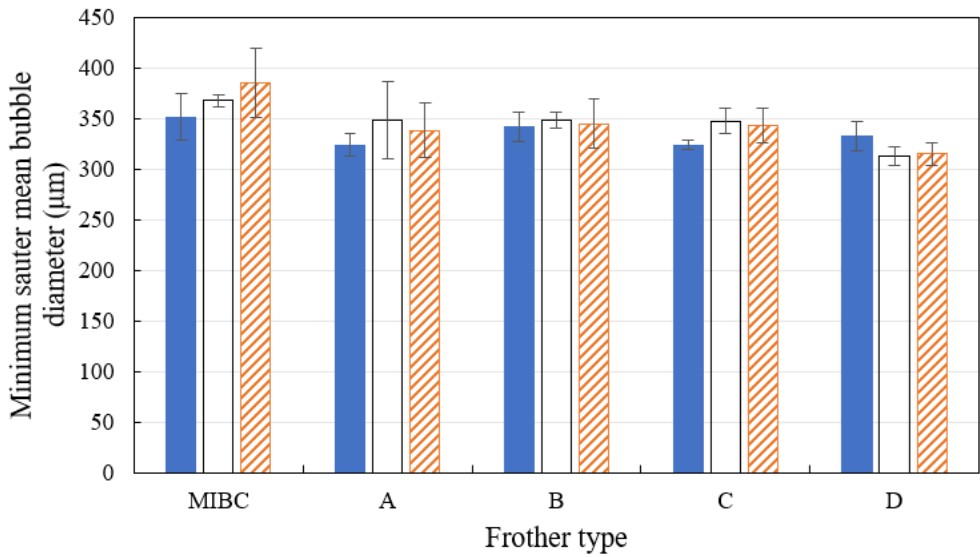

**Figure 13.** Minimum bubble size obtained at different frother types and water salinities conditions. The error bars represent the 95% confidence intervals of the mean values.

The results clearly show that the minimum measured Sauter mean bubble diameter was within the range between 320–370 μm. The minimum bubble size was found to be largely independent of the solution chemistry (both frother type and salinity level). The small variations in the minimum bubble size measurements fit within the margin of experimental error.

This result is expected, as above the CCC, the size of the bubbles is expected to be dominated by the bubble sparging mechanism and/or the cell hydrodynamic conditions. All the hydrodynamic variables (such as $J_g$, cell volume, and stirring rate) were kept constant throughout the tests, resulting in consistent minimum bubble size.

### 4. Conclusions

A novel bubble size measurement tool, the Mettler Toledo ParticleView[TM] Measurement probe (PVM), was tested in a lab-scale bubble generation system and was compared with a conventional tool, the Anglo Platinum Bubble Sizer (APBS), under spherical bubble generation regimes. Results show that PVM and APBS probes both provide similar Sauter mean bubble diameter measurements. The difference between the two measurements was small, but statistically significant. Furthermore, the two probes present strengths in different areas. PVM probe provides a clearer vision of micro-field views and needs more photos to be taken to ensure a minimum bubble count. It is more compact and light, so it is easy to set up and use in a laboratory environment. Conversely, the APBS did not detect

small bubbles less than 200 μm in size, and this might be because of the counter-current flow of water going down through the sampling tube. More work is required to further confirm on this. APBS is much larger in size and heavier to move around as compared with PVM. It is suitable for both lab and site work, however, it is less convenient to install in a lab context as compared to PVM.

The PVM probe was then used to characterize five frother formulations for a coal mine in Queensland, Australia. Bubble size measurement was carried out for each frother at progressively increasing concentrations and various background water salinity levels. Sauter mean bubble diameter was calculated and plotted against frother dosage, from which the Critical Coalescence Concentration (CCC) was determined and compared.

Results show that the PVM gave a good measurement of bubble size that allowed the performances of the different frother formulations to be clearly distinguished. The image processing algorithm worked well for spherical regimes under the current study, and more complex algorithms [23] will be further explored in the future for characterization of bubbles under ellipsoidal or churn-turbulent conditions. The technique also further confirmed the different extent to which the frother formulations are sensitive to the changes in process water within the range relevant to industrial practice. Frother D is the most effective frother, requiring less dosage to achieve the same frothing characteristics. The plant will need to use less frother during periods of the high salinity of process water to achieve the minimum Sauter mean bubble size.

**Author Contributions:** Conceptualization, E.F. and J.W.; methodology, E.F., J.W. and G.F.; software, G.F.; formal analysis, E.F.; investigation, E.F. and J.W.; writing—original draft preparation, J.W.; writing—review and editing, E.F. and G.F. All authors have read and agreed to the published version of the manuscript.

**Funding:** This research was funded by Anglo American.

**Data Availability Statement:** Restrictions apply to the availability of these data. Data was obtained using funding from Anglo American, and are available from the corresponding author with the permission of Anglo American.

**Acknowledgments:** The authors would like to acknowledge Anglo American for the financial support to the current study, as well as Mathew Merryweather from Anglo American for providing support. The authors also acknowledge Ding Sheng Kaw, Joyce Siong, Konuray Demir, Stephen Dickinson, Dylan Carr, and other JKMRC staff for their assistance in this work.

**Conflicts of Interest:** The authors declare no conflict of interest.

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
