# Peer review of "Frother Characterization Using a Novel Bubble Size Measurement Technique"

_applsci, doi:10.3390/app12020750_

Round 1
Reviewer 1 Report
This paper presents a new probe to estimate bubble size in flotation, below 2 mm. The technique seems atractive for bubbles smaller than 0.9 mm. However, this new measurement tool presents some issues.
General Comments:
- The authors do not indicate how they correct the variable depth of field. This is a critical point to estimate bubble size.
- The technique may be applicable (with some corrections) in spherical regimes. The authors did not discuss this point in the literature review, which is the baseline to use the proposed method. More interesting than measuring under spherical regime is to conduct characterizations under ellipsoidal or churn-turbulent conditions. Conventional Bubble viewers + circle detection are good enough to characterize spherical bubbles. It would be interesting to evaluate the technique at higher JGs.
- The Hough Transformation has been previously used to estimate bubble size in flotation. I suggest citing those references. Again, if the flotation machine were operated transitioning to an ellipsoidal regime some incomplete bubbles may be estimated as spheres, depending on the threshold used in the circle detection (which was not resported in the current draft). Please evaluate a possible bias (again at higher JG).
- The comparison with conventional bubble viewers for dB < 0.2 mm is incorrect. To improve the bubble detection with an APBS, only a change in the camera resolution or lens will enhance the characterization.
- What is the change in dB after the pressure correction? Is it significant regarding the resolution or measurement error? Is it more significant than distorsions due to variable depth of field?
- It would be interesting to see a picture of a lab rig.
Author Response
Reviewer #1:
This paper presents a new probe to estimate bubble size in flotation, below 2 mm. The technique seems atractive for bubbles smaller than 0.9 mm. However, this new measurement tool presents some issues.
General Comments:
- The authors do not indicate how they correct the variable depth of field. This is a critical point to estimate bubble size.
Answer: The authors appreciate this comment. For the consideration of the variable depth of field, here are the cases for APBS and PVM, respectively.
APBS – APBS is a mature product and technique well recognized by the flotation industry. In the current study, we followed exactly the procedure to conduct experiments and data analysis with using its own software without further adjusting any parameters. It is assumed that the variable depth of field is already considered at the time of the development of this product.
PVM – PVM has a very small field of view, 0.89 mm × 1.3 mm.The number of bubbles captured in one image ranges from only one to just a couple. Given that bubbles are densely populated in the bubble generation system, it is assumed that the bubbles captured in the images are the ones sitting closely to the camera lens. In this way, the effect of the variable depth of field can be neglected.
- The technique may be applicable (with some corrections) in spherical regimes. The authors did not discuss this point in the literature review, which is the baseline to use the proposed method. More interesting than measuring under spherical regime is to conduct characterizations under ellipsoidal or churn-turbulent conditions. Conventional Bubble viewers + circle detection are good enough to characterize spherical bubbles. It would be interesting to evaluate the technique at higher JGs.
Answer: We have followed the reviewer’s suggestion exactly and added the information in terms of the potential use of the novel technique in spherical bubble detection situations. The application of the technique in ellipsoidal or chun-turbulent regimes is to be explored in the future work.
- The Hough Transformation has been previously used to estimate bubble size in flotation. I suggest citing those references. Again, if the flotation machine were operated transitioning to an ellipsoidal regime some incomplete bubbles may be estimated as spheres, depending on the threshold used in the circle detection (which was not resported in the current draft). Please evaluate a possible bias (again at higher JG).
Answer: We have followed the reviewer’s suggestion exactly and cited the previous work regarding the use of Hough Transformation in Circular bubble detection. Also, we have followed the reviewer’s suggestion and further clarified that the current study only focused on the spherical regimes and that the application of the technique in ellipsoidal or chun-turbulent regimes is to be explored in the future work.
- The comparison with conventional bubble viewers for dB < 0.2 mm is incorrect. To improve the bubble detection with an APBS, only a change in the camera resolution or lens will enhance the characterization.
Answer: We agree with the reviewer that a change in the camera lens and the corresponding resolution might enhance the characterization. What we observed in the current study is that small bubbles are captured in the images together with other big bubbles, however, the detection of these bubbles in the following image analysis is facing some issues. The field of view of APBS is 68.5mm by 91 mm, in order to detect the bubbles <0.2 mm, the sensitivity of the software needs to be adjusted. At this stage, incorrect captures of non-bubbles as small bubbles were already observed. Further improving the sensitivity would inevitably bring in more fault detection of non-bubble objects. This is the reason that the paper reached a conclusion that it might not be easy for APBS to capture really small bubbles less than 0.2 mm down to ~20 microns.
- What is the change in dB after the pressure correction? Is it significant regarding the resolution or measurement error? Is it more significant than distortions due to variable depth of field?
Answer: The pressure correction factor is 0.97. So the difference is not significant when comparing the bubble size before and after pressure correction. The comment regarding the distortions due to variable depth of field had been addressed in the first question field.
- It would be interesting to see a picture of a lab rig.
Answer: We have followed the reviewer’s comment exactly and added a picture showing the lab rig in the manuscript.
Reviewer 2 Report
In this paper, the authors evaluated the Metter Toledo Particle View Measurement probe for in-situ bubble size measurements (PVM). The authors made a comparative analysis of bubble size distributions and the Sauter mean bubble diameter obtained with this technique and with the conventional Anglo Platinum Bubble Sizer (StoneThreeTM) (APBS) technique, using MIBC as frother. Subsequently, they determine the CCC and the minimum Sauter mean bubble diameter for different types of frothers and different saline medias. The results obtained showed that PVM and APBS probes both provide similar Sauter mean bubble diameter measurements and PVM gave a good measurement of bubble size of the different frother formulations in different saline medias.
The paper is suitable to be published in the Applied Sciences Journal, however several corrections should be made:
- In general, the authors should correct the numbering of the entire document because they are not correct.
- In page 2, section introduction, lines 5-7. Please, add a reference after this idea “However, excess frother use causes problems in downstream thickeners and recycle water pumping systems due to over-stable froth”.
- In page 3, section introduction, please eliminate the title “objectives”.
- In page 4, section Frothers. The Table is not named in the previous paragraph.
- Please indicate the values of the ionic strength of this synthetic process water in Table 1.
- In relation to the report in the Figure 2. Please indicate the following
- What is the importance of this information in this study? Please clarify!
- What does each point shown on the graph correspond to? (plant and laboratory)
- Please improve this section, it is necessary to explain the methodology when APBS and PVM were used, in order to evidence that the installation using the PVM is easier than APBS technique.
- Please improve the image showed in the figure 6a.
- Figure 8. Please add the error bars in the curve of APBS
- In page 9, when the results shown in the figure 8 are analyzed. The authors indicate that “0.57% of bubbles cap-tured with the APBS probe were larger than 2 mm and were not detected with the PVM probe.” However, the results > 2mm are the same in both cases. How do you indicate this percentages? Please clarify the idea.
- In the section of the two phase frother characterization. Why the comparative results obtained with PVM an APBS were not included? This in order to evidence that the results are similar with both techniques.
- Figure 9, Please improve the resolution or the quality of this figure.
Author Response
Reviewer #2:
In this paper, the authors evaluated the Metter Toledo Particle View Measurement probe for in-situ bubble size measurements (PVM). The authors made a comparative analysis of bubble size distributions and the Sauter mean bubble diameter obtained with this technique and with the conventional Anglo Platinum Bubble Sizer (StoneThreeTM) (APBS) technique, using MIBC as frother. Subsequently, they determine the CCC and the minimum Sauter mean bubble diameter for different types of frothers and different saline medias. The results obtained showed that PVM and APBS probes both provide similar Sauter mean bubble diameter measurements and PVM gave a good measurement of bubble size of the different frother formulations in different saline medias.
The paper is suitable to be published in the Applied Sciences Journal, however several corrections should be made:
- In general, the authors should correct the numbering of the entire document because they are not correct.
Answer: We have followed this suggestion and corrected all the numbering throughout the document.
- In page 2, section introduction, lines 5-7. Please, add a reference after this idea “However, excess frother use causes problems in downstream thickeners and recycle water pumping systems due to over-stable froth”.
Answer: We followed the reviewer’s suggestion and added a reference in this part. Also we have further clarified the effect of over-dosing of frothers. The description has been changed to the following:
However, excess frother use might eventually destabilize the froth and lead to a slower flotation rate [Fuerstenau, 2011 #39].
- In page 3, section introduction, please eliminate the title “objectives”.
Answer: We have followed this suggestion. The title of “objectives” had been deleted and this section had been combined to the Introduction section.
- In page 4, section Frothers. The Table is not named in the previous paragraph.
Answer: We have followed this suggestion and added a sentence at the end of the paragraph – ‘Details of the five frothers under study are given in Table 1’.
- Please indicate the values of the ionic strength of this synthetic process water in Table 1.
Answer: We had a second thought on the structure of materials used in the current study, and decided to keep the present form. The reasons for this are followed. Firstly, the addition of frothers into the synthetic process water system was at ppm levels and it was assumed that this would only affect the ionic strength of the system marginally. Secondly, the frothers were grouped together and shown in Table 1 to give the audience a better understanding of the difference in the frothers used in the current study, while the ionic strength of the synthetic process water was compared with the plant water and shown in Figure 2. In this way, the audience could locate the information they are after quickly. Therefore, we have kept this format to the original.
- In relation to the report in the Figure 2. Please indicate the following
- What is the importance of this information in this study? Please clarify!
- What does each point shown on the graph correspond to? (plant and laboratory)
Answer: The authors appreciate the comment. This information is important because it illustrates the natural range of the variability of process water salinity at a real operation. Therefore, the range of ionic strengths chosen for these experiments accurately represents the rage of a real world system. Figure 2 shows the comparison of the process water characteristics achieved at plant and laboratory, and the results showed that the synthetic process water made at lab was close to that obtained from plant and thus was a good representation of the real situation.
We appreciate the second comment and have further clarified in the manuscript that the points represent the process water at different salt concentrations.
- Please improve this section, it is necessary to explain the methodology when APBS and PVM were used, in order to evidence that the installation using the PVM is easier than APBS technique.
Answer: We followed this suggestion exactly and have added a section in the Materials and Experimental Methods part showing the lab-scale experimental set-up for the PVM and APBS installations, respectively.
- Please improve the image showed in the figure 6a.
Answer: We have followed this suggestion and improved the image shown in Figure 6a.
- Figure 8. Please add the error bars in the curve of APBS
Answer: The error bars of APBS curve had already been added on to the figure. The reason that they could hardly be seen is because that the error bars were small.
- In page 9, when the results shown in the figure 8 are analyzed. The authors indicate that “0.57% of bubbles captured with the APBS probe were larger than 2 mm and were not detected with the PVM probe.” However, the results > 2mm are the same in both cases. How do you indicate this percentages? Please clarify the idea.
Answer: The authors appreciate this comment. In terms of how the value 0.57% was calculated, we counted the number of all of the bubbles detected, as well as that of bubbles larger than 2mm, and calculated the corresponding percentage of bubbles >2mm as 0.57%. Figure 8 shows the information of big bubbles of diameters 2mm to 10mm. The cumulative percentage value of this range makes up the 0.57% value, which is the reason that it was not as easy to distinguish as the case found in the small bubble range (<177 microns).
- In the section of the two phase frother characterization. Why the comparative results obtained with PVM an APBS were not included? This in order to evidence that the results are similar with both techniques.
Answer: Firstly, the same salts were used to make up the synthetic process water, and the same frother (MIBC) was used in the PVM-APBS comparison part as compared with the tests conducted for the frother characterization work. The only difference between the two systems was the rig used to generate bubbles, a column with a sparger placed on its bottom for the former and a bottom motor driven column for the latter. It is assumed that the results obtained in one system applies to another.
Secondly, it is not feasible to install APBS on to the rig that was used for the frother characterization work. This column that was used to generate bubbles was driven by a motorand thus can only be sat on a bench, which left very limited space for APBS to be installed. The lab height is about 2.5m long and APBS is about 2m long (from top of the rig to the endpoint of the sampling tube). This is also one of the conclusions that this paper reaches, that is, PVM probe is much easier to use than APBS in the lab work context.
- Figure 9, Please improve the resolution or the quality of this figure.
Answer: The authors appreciate this comment and have improved the resolution of Figure 9.
Reviewer 3 Report
Interesting work on in-situ bubble size analysis in flotation cells although as it stands it feels closer to a technical note since it is unclear what the novelty is other than the technique. Several issues for improving the manuscript are identified.
- It feels in places a bit of an advertisement for Mettler Toledo. Is this really necessary in the asbtract/conclusions? Also, what about the limitations of this technique?
- The description of the APBS should be expanded and revised for accuracy.
- No details are given for the fps used for the images obtained during the APBS tests, which should be the same as for the PVM for a valid comparison.
- Why not use the dedicated algorithm developed for the PVM on the images obtained with the APBS? This would avoid hypothesising about the reasons why the APBS software might not be detecting smaller bubbles.
- The impact of salts on bubble size has been studied in the past. I wonder whether that should be the focus of this work.
- Frother A, B, C and D must be detailed in this work. Incidentally, the reference given for details is shown as an "Error" in the document.
- Why such low Jgs were chosen for the tests?
- Further validations would be ideal - at present the authors compare their measurements with an established method (APBS) for a single condition. Also, were there repeats for the APBS data (as no confidence intervals are shown)?
- Many other figures are missing data on repeatability/experimental error.
- I believe there is risk with the PVM of very small, slower bubbles, being counted in consecutive images and thus biasing the results?
- More details on the setup for both techniques and image analysis for the APBS software, are needed.
- The conclusions mention the PVM is "more compact and light" but this seems out of place here.
- When stating that the APBS does not detect smaller bubbles...does this mean the software doesn't rather than the equipment itself? Much clarity is needed here. It is felt there is too much hypothesising about the effect of counter-current wash-water.
- Also in the conclusions, the statements about "good measurements of bubble size" are subjective and to this reviewer's opinion, non justified by the work presented.
- The technique has not "demonstrated the different extent to which the frother formulations are sensitive to the changes in process water". This is well known.
- The inability of the PVM to capture the larger bubbles due to the reduced field of view is a real issue, as those bubbles are most probably under-represented in the APBS measurements anyway, then compounding something that is already an issue in existing methods.
I encourage the authors to address the issues above.
Author Response
Reviewer #3:
Interesting work on in-situ bubble size analysis in flotation cells although as it stands it feels closer to a technical note since it is unclear what the novelty is other than the technique. Several issues for improving the manuscript are identified.
- It feels in places a bit of an advertisement for Mettler Toledo. Is this really necessary in the asbtract/conclusions? Also, what about the limitations of this technique?
Answer: The authors appreciate this comment and have removed Mettler Toledo’s name in the abstract. The authors have no conflict of interests with Mettler Toledo company. The objective of the current work is to introduce an in-situ technique easier for lab work bubble size measurement.
The limitations of this technique include 1) hard to use in a plant test context; 2) difficult to capture big bubbles because of its small field of view. These limitations have been discussed in the body part and were also summarized in the conclusion part.
- The description of the APBS should be expanded and revised for accuracy.
Answer: We have followed the suggestion exactly and kept one individual section for APBS introduction, and further checked the accuracy of the description of the technique.
- No details are given for the fps used for the images obtained during the APBS tests, which should be the same as for the PVM for a valid comparison.
Answer: The authors appreciate this comment. The PVM image taken was set as 1 fps. As for APBS, we controlled the image taken manually, and took one image per second as well. We have followed this suggestion exactly and added this information in the manuscript.
- Why not use the dedicated algorithm developed for the PVM on the images obtained with the APBS? This would avoid hypothesising about the reasons why the APBS software might not be detecting smaller bubbles.
Answer: The authors appreciate this comment. We actually tried to use the same algorithm for both sets of pictures obtained (APBS software for pictures obtained with PVM, and vice versa algorithm developed for PVM for pictures obtained with APBS). However, the results were not making any sense.
In terms of the small bubble detection with APBS, the field of view of APBS is in the cm range, in order to capture the bubbles ~ 20 microns, some sensitivity parameters of APBS software needs to be adjusted. However, at this stage, incorrect captures of non-bubbles as small bubbles were already observed. Further improving the sensitivity would inevitably bring in more fault detection of non-bubble objects. This is the reason that the paper reached a conclusion that it might not be easy for APBS to capture really small bubbles ~20 microns.
- The impact of salts on bubble size has been studied in the past. I wonder whether that should be the focus of this work.
Answer: The authors appreciate this comment and agree with the reviewer’s idea. The impact of salts on bubble size has long been studied, and this is not the focus of the current study. The main focus of the current study is to 1) compare this new in-situ tool with conventional recognized technique APBS, and to 2) apply it in a real system to solve a problem. In order to achieve the latter, salts were used to make synthetic process water, mimicking the real situation at plant, and based on that, this new in-situ tool was tested.
- Frother A, B, C and D must be detailed in this work. Incidentally, the reference given for details is shown as an "Error" in the document.
Answer: Details of Frothers A, B, C, and D have been provided in the work, including the suppling company and main components of each frother. The reference was double checked to make sure correct information was shown in the document.
- Why such low Jgs were chosen for the tests?
Answer: The authors appreciate the reviewer’s comment. Yes, the Jgs used in the current study (0.25cm/s) is low, but still within the range operated at plant. The cross-sectional area of the rigs were big (20cm side length), for a fixed total air flowrate (6L/min), it resulted in a relatively low Jg value.
- Further validations would be ideal - at present the authors compare their measurements with an established method (APBS) for a single condition. Also, were there repeats for the APBS data (as no confidence intervals are shown)?
Answer: The authors appreciate the reviewer’s comment. Yes, repeat tests were conducted for APBS as well, and the error bars of APBS curve were also shown on the figure. The reason that they could hardly be seen is because that the error bars were small.
- Many other figures are missing data on repeatability/experimental error.
Answer: The authors appreciate the reviewer’s comment and agree with the reviewer that it would be better if all figures show error bars. For the current work, it is a bit difficult to achieve that. There are five frother under study. For each frother, three ionic strength of process water were tested. For each process water condition, six frother concentrations were tested. This results in a total of 90 tests. As a result, only the comparison of PVM and APBS tests were repeated, and error bars were reported. Given that the errors turned out to be small, and that the same techniques were used in the current study, it is assumed that the errors for other tests stood at a similar level.
- I believe there is risk with the PVM of very small, slower bubbles, being counted in consecutive images and thus biasing the results?
Answer: The authors had the same concern, so the frequency of image taken was set as one image per second. The images were inspected before image analysis, and no repeated images were found from two images in a row.
- More details on the setup for both techniques and image analysis for the APBS software, are needed.
Answer: We have followed this suggestion exactly and added a section in the Materials and Experimental Methods part showing the lab-scale experimental set-up for the PVM and APBS installations, respectively.
APBS is a commercial product, and it is not easy to get further details in terms of the image analysis procedure.
- The conclusions mention the PVM is "more compact and light" but this seems out of place here.
Answer: The authors appreciate the reviewer’s comment and have added a figure showing the lab-scale experimental set-up for the PVM and APBS installations. Together with the dimension comparison of the two tools made in Section 3.1, it reached the conclusion that PVM is “more compact and light”.
- When stating that the APBS does not detect smaller bubbles...does this mean the software doesn't rather than the equipment itself? Much clarity is needed here. It is felt there is too much hypothesising about the effect of counter-current wash-water.
Answer: The detection of smaller bubbles (<0.2 mm) with using APBS was not as good as that obtained with PVM. The field of view of APBS is 68.5mm by 91 mm, in order to detect the bubbles <0.2 mm, the sensitivity of the software needs to be adjusted. At this stage, incorrect captures of non-bubbles as small bubbles were already observed. Further improving the sensitivity would inevitably worsen the accuracy of the results. This is the reason that the paper reached a conclusion that it might not be easy for APBS to capture really small bubbles less than 0.2 mm down to ~20 microns.Instead, it is very capable of coping with bubbles >0.2mm up to a few cm.
In terms of the potential effect of counter-current wash-water, we agree that there is much hypothesising involved. More work is needed to confirm this effect. Thus we have added this in the manuscript.
- Also in the conclusions, the statements about "good measurements of bubble size" are subjective and to this reviewer's opinion, non justified by the work presented.
Answer: The authors appreciate this comment and have made some revisions in this paragraph accordingly. The Sauter mean bubble diameter obtained with using PVM was comparable with that obtained with using the well-recognized tool APBS, so it is assumed that the data obtained with PVM would give a good representation of the real bubble size in the system. We have deleted the subjective word ‘good’ from the text.
- The technique has not "demonstrated the different extent to which the frother formulations are sensitive to the changes in process water". This is well known.
Answer: The authors appreciate this comment and have rephrased the sentence, “further confirmed the different extent to which the frother formulations are sensitive to the changes in process water”.
- The inability of the PVM to capture the larger bubbles due to the reduced field of view is a real issue, as those bubbles are most probably under-represented in the APBS measurements anyway, then compounding something that is already an issue in existing methods.
Answer: The authors appreciate this comment. It is true that it is a real issue and limiting point to use PVM to capture larger bubbles. It showed stronger capability on the micron-level range. On the contrary, the field of view of APBS is much larger, in the cm-10s cm range, and provides a good solution for big bubble capture. This has been discussed in the manuscript, showing the strengths and limitations of each technique.
- I encourage the authors to address the issues above.
Answer: The authors appreciate all the comments and suggestions and have thus addressed all issues mentioned above accordingly.
We really appreciate the constructive and thoughtful comments from these Reviewers.
Round 2
Reviewer 1 Report
The authors should be congratulated for this innovative piece of research.
Reviewer 2 Report
The authors made most of the requested changes
Reviewer 3 Report
Other than the data on repeatability, which is still missing, changes has been made that have improved the manuscript. If the Editor considers that is sufficient, I have no further objections.